# New Fusarin Derivatives from the Marine Algicolous Fungus *Penicillium steckii* SCSIO41040

**DOI:** 10.3390/md21100532

**Published:** 2023-10-12

**Authors:** Yingying Song, Jianglian She, Weihao Chen, Jiamin Wang, Yanhui Tan, Xiaoyan Pang, Xuefeng Zhou, Junfeng Wang, Yonghong Liu

**Affiliations:** 1CAS Key Laboratory of Tropical Marine Bio-Resources and Ecology, Guangdong Key Laboratory of Marine Materia Medica, South China Sea Institute of Oceanology, Chinese Academy of Sciences, Guangzhou 510301, China; songyingying19@mails.ucas.ac.cn (Y.S.); shejianglian20@mails.ucas.ac.cn (J.S.); chenweihao17@mails.ucas.ac.cn (W.C.); wangjiamin20@mails.ucas.ac.cn (J.W.); xypang@scsio.ac.cn (X.P.); xfzhou@scsio.ac.cn (X.Z.); 2University of Chinese Academy of Sciences, Beijing 100049, China; 3State Key Laboratory for Chemistry and Molecular Engineering of Medicinal Resources, School of Chemistry and Pharmaceutical Sciences, Guangxi Normal University, Guilin 541004, China; tyh533@126.com

**Keywords:** fusarin derivatives, *Penicillium steckii*, marine algicolous fungus

## Abstract

Five new fusarin derivatives, steckfusarins A–E (**1**–**5**), and two known natural products (**6**, **7**), were isolated and identified from the marine algicolous fungus *Penicillium steckii* SCSIO 41040. The new compounds, including absolute configurations, were determined by spectroscopic analyses and calculated electronic circular dichroism (ECD). All new compounds were evaluated for their antioxidant, antibacterial, antifungal, antiviral, cytotoxic, anti-inflammatory, antioxidant, cholesterol-lowering, acetyl cholinesterase (AChE) enzyme and 6-phosphofructo-2-kinase (PFKFB3) and phosphatidylinositol-3-kinase (PI3K) inhibitory activities. The biological evaluation results revealed that compound **1** exhibited radical scavenging activity against 2,2-diphenyl-1-picrylhydrazylhydrate (DPPH), with an IC_50_ value of 74.5 µg/mL. In addition, compound **1** also showed weak anti-inflammatory activity at a concentration of 20 µM.

## 1. Introduction

Based on genome sequencing and mining and antimicrobial screening of crude extracts, *P. steckii* has shown strong activity against some pathogenic bacteria [1]. It has been recently reported that the secondary metabolites of *P. steckii* were mainly tanzawaic acids and alkaloids that demonstrated a broad range of significant biological activities, such as antimicrobial, anti-inflammatory and lipid-lowering activities [2,3,4,5,6].

Fusarins are a class of mycotoxins characterized by a 2-pyrrolidone moiety and a triene side chain of twelve carbon atoms [7] that are usually isolated from *Penicillium* sp. [8]. The compounds were biosynthesized by a hybrid polyketide synthase-nonribosomal peptide synthetase (PKS-NRPS) [9]. In the human neuroblastoma cell line SH-SY5Y, fusarin derivatives were first proven to effectively promote neurite growth and block the cell cycle in the G_0_/G_1_ phase [10]. Fusarin derivatives inhibit mammalian DNA polymerases and human DNA topoisomerase II in vitro [11]. Additionally, they induce the formation of 6-thioguanine-resistant mutants in V79 cells, as well as the asynchronous replication of polyoma DNA sequences subsequent to the presence of a microsomal activation system [12,13]. Fusarins have been reported to be unstable after prolonged exposure to ultraviolet light and elevated temperatures [14]; additionally, light can induce the rearrangement of fusarins, leading to the simultaneous appearance of fusarin analogues [15].

During our continuous study exploring new bioactive natural products from marine microorganisms [16,17], five new fusarin derivatives (**1**–**5**) (Appendix A) and two known natural products (**6**, **7**) (Figure 1) were obtained from the fungus *Penicillium steckii* SCSIO 41040, which was isolated from a green algae *Botryocladia* sp. of the South China Sea. Herein, we describe the fermentation, isolation, structural determination and biological activities of these compounds.

## 2. Results and Discussion

Compound **1** was obtained as a yellow oil, and the molecular formula C_21_H_27_NO_6_ was assigned to it based on its HRESIMS (*m*/*z* 390.1912 [M + H]^+^, calcd for C_21_H_28_NO_6_^+^, 389.1911), which required nine degrees of unsaturation. The ^1^H NMR and ^13^C NMR data (Table 1) indicated five olefinic methines (*δ*_C/H_ 138.5/6.80, CH-2; 123.5/5.94, CH-4; 134.9/6.25, CH-6; 128.4/5.73, CH-7; 148.0/6.86, CH-10), five methyls (*δ*_C/H_ 15.6/1.66, CH_3_-1; 18.7/1.39, CH_3_-18; 14.2/1.56, CH_3_-20; 10.8/1.75, CH_3_-21; 49.5/3.26, CH_3_-22), two methylenes (*δ*_C/H_ 31.0/2.26, CH_2_-8; 28.8/2.42, CH_2_-9), one oxygenated methine (*δ*_C/H_ 62.4/4.28, CH-14) and eight quaternary carbonyls (*δ*_C_ 131.0, C-3; 136.7, C-5; 137.5, C-11; 190.0, C-12; 62.2, C-13; 87.4, C-15; 169.9, C-17; 167.9, C-19). Seven degrees of unsaturation could be accounted for by the presence of three carbonyl signals (a ketone resonance (*δ*_C_ 190.0), an amide (*δ*_C_ 169.9) and a carboxyl (*δ*_C_ 167.9)) and the eight *sp*^2^ carbon signals of four olefinic bonds. Therefore, the remaining unsaturation indicated the presence of two rings. HMBC correlations from H-14 to C-15 and NH-16 to C-13/C-14/C-15/C-17 established the presence of the 2-pyrrolidone ring structure. The highly deshielded chemical shift of C-13 and CH-14 inferred that both were oxygenated, which prompted us to speculate on the existence of an epoxide ring in the 2-pyrrolidone ring. Additionally, the location of the methyl and methoxyl groups were determined via HMBC correlations from H_3_-18 to C-14/C-15 and H_3_-22 to C-15. According to the 2D NMR spectra, a 2-ethylidene-4,10-dimethyl-11-oxoundeca-3,5,9-trienoic acid side chain was located at C-13 of the 2-pyrrolidone ring, supported by COSY correlations of H_3_-1/H-2 and H-6/H-7/H_2_-8/H_2_-9/H-10 and HMBC correlations from H_3_-1 to C-3, H-2 to C-19, H-4 to C-2/C-6, H_3_-20 to C-4/C-5 and H_3_-21 to C-10/C-11/C-12. The *E*-configuration of the ∆^6(7)^ double bond was determined by the coupling constant (*J* = 15.6 Hz). NOESY correlations indicated an *E*-configuration for the other double bonds in the side chains of H_3_-1/H-4, H-4/H-6, H_2_-9/H_3_-21 and the cofacial orientation of H_3_-18/H-14. The presence of an epoxide ring indicated the homoconfiguration of H-13/H-14. The absolute configurations of compound **1** were determined by ECD calculations. The experimental ECD spectra were compared with two calculated spectra, (13*R*, 14*R*, 15*R*)-**1** and (13*S*, 14*R*, 15*R*)-**1**. The results showed that the experimental ECD spectra of **1** matched well with the calculated spectrum for (13*R*, 14*R*, 15*R*)-**1**, confirming the absolute configurations of **1** as depicted (Figure 2).

Compound **2** was isolated as a yellow oil. The HRESIMS (*m*/*z* 347.1609 [M − H]^−^, calcd for C_20_H_24_NO_6_^−^, 374.1609) data suggested a molecular formula of C_20_H_25_NO_6_, revealing nine degrees of unsaturation. The NMR signals (Table 1) of **2** resembled those of **1**, except for the absence of a methoxyl group at *δ*_C/H_ 49.5/3.26. Large coupling (*J* = 15.6 Hz) between H-6 (*δ*_H_ 6.25) and H-7 (*δ*_H_ 5.74) showed that the ∆^6(7)^ olefinic bond was in *E*-configuration. NOESY data of H_3_-1 (*δ*_H_ 1.65)/H-4 (*δ*_H_ 5.93), H-4 /H-6 and H_2_-9 (*δ*_H_ 2.40)/H_3_-21 (*δ*_H_ 1.75), indicated that the ∆^2(3)^, ∆^4(5)^ and ∆^10(11)^ olefinic bonds were all in *E*-configuration. The NOESY spectrum of **2** showed key correlations of H_3_-18/H-14, indicating that they were co-facial. The high similarity between the calculated ECD curve of (13*R*, 14*R*, 15*R*)-**2** and its experimental curve unambiguously confirmed the absolute configuration of **2** (Figure 2) [18,19,20].

Compound **3** was obtained as a pink oil. Its molecular formula of C_25_H_33_NO_8_ was determined via HRESIMS (*m*/*z* 476.2290 [M + H]^+^, calcd for C_25_H_34_NO_8_^+^, 476.2279), which required ten degrees of unsaturation. Proton network systems of H-13 (*δ*_H_ 4.93)/H-14 (*δ*_H_ 4.33) and H_2_-23 (*δ*_H_ 3.51, 3.67)/H-24 (*δ*_H_ 4.16)/H_2_-25 (*δ*_H_ 3.34, 3.99) were detected in the COSY spectrum. Moreover, HMBC correlations from H-13 to C-17 (*δ*_C_ 170.5), NH-16 (*δ*_H_ 8.89) to C-13 (*δ*_C_ 48.8)/C-14 (*δ*_C_ 77.2)/C-15 (*δ*_C_ 82.9)/C-17, H_3_-18 (*δ*_H_ 1.48) to C-14/C-15 and H_2_-23 to C-15 established the presence of a glycerol moiety and the 2-pyrrolidone ring structure, which were connected via ether linkages to form a 8a-methyl-2-oxohexahydro-1*H*,5*H*-[1,4]dioxepino [2,3-*b*]pyrrol-6-yl acetate ring system (Figure 3). A detailed analysis of its NMR spectroscopic features implied that it was closely structurally related to dothilactaene B [15]. A set of signals at CH_3_-22 and C-26 (Table 1 and Table 2), which were not present in the ^1^H and ^13^C NMR spectra of dothilactaene B, were observed in the spectrum of **3**. Meanwhile, compound **3** lacked a methoxy group at *δ*_C_/_H_ 51.9/3.55. Through the analyses of the HMBC spectrum, the acetic acid group was confirmed by the HMBC correlations from H_3_-22 (*δ*_H_ 1.98) to C-26 (*δ*_C_ 170.8) and H_2_-25 to C-26. The *E*-configuration of the ∆ ^6(7)^ double bond was determined by the coupling constant (*J* = 15.6 Hz). NMR data analyses of **3** showed a similar multiplicity pattern in the glycerol moiety, indicating that they are two components and have the same plane structure. The resonances of the glycerol moiety suggested them to be diastereoisomers presenting in a ratio of 3:1 (**3a**:**3b**), which may be formed by Michael addition [21]. NOESY correlations (Figure 4) of **3a** indicated the *E*-configuration for the other double bonds in the side chain and the cofacial orientation of H-13, H-14, CH_3_-18 and H-24. The NOESY spectrum of **3b** was the same as **3a**, except the signal of H-24 was the opposite. These data permitted assignment of the (13*R**, 14*R**, 15*R**, 24*R**)-**3a** and (13*R**, 14*R**, 15*R**, 24*S**)-**3b** relative configurations [15,19].

Compound **4** was obtained as a yellow oil. Its HRESIMS ([M + H]^+^, 346.2016; calcd for C_20_H_28_NO_4_^+^, 346.2013) data were in agreement with the molecular formula C_20_H_27_NO_4_. Its ^1^H NMR and ^13^C NMR data (Table 3) closely resembled the data for **2**, except for the absence of the hydroxyl group and an epoxy system at the 2-pyrrolidone ring structure. To data, all fusarin derivatives reported from natural sources have had a free hydroxyl group at C-15; however, compound **4** has no such substituent [22,23,24]. The relative configuration of **4** was confirmed by its NOESY correlations and coupling constants (Figure 4). The NOESY correlations of H_3_-1 (*δ*_H_ 1.64)/H-4 (*δ*_H_ 5.95), H-4/H-6 (*δ*_H_ 6.26), H_2_-9 (*δ*_H_ 2.38)/H_3_-21 (*δ*_H_ 1.72) and the large coupling constant of H-6/H-7 (*J* = 15.6 Hz) suggested the *E* geometry of all double bonds. Each pair of ^1^H and ^13^C NMR signals showed a similar multiplicity pattern, suggesting that compound **4** was an inseparable mixture of two geometric isomers presenting in a ratio of 1:1 (**4a**:**4b**). Even under mild conditions, the stereo center of C-15 was easy to be epimerized through ring opening [19,20]. Comparing **4a** and **4b** (Table 2), the main difference occurred at the configuration at C-13, C-15 and C-18, which was proven by the carbon chemical shifts of C-13 (∆*δ*_C_ 1.1 ppm), C-15 (∆*δ*_C_ 1.1 ppm) and C-18 (∆*δ*_C_ 0.2 ppm). The NOESY spectrum of **4a** showed cross-peaks between H-13 (*δ*_H_ 4.33) and H_3_-18 (*δ*_H_ 1.11) that indicated H-13 and H-15 (*δ*_H_ 1.48) were on the opposite side, whereas no relevant signal between H-13 and H_3_-18 was found in **4b**. Accordingly, the relative configurations of **4a** and **4b** could be assigned as (13*R**, 15*R**)-**4a** and (13*R**, 15*S**)-**4b**.

Compound **5** was obtained as a yellow oil. It showed a [M + H]^+^ ion peak at *m*/*z* 265.1431 in the positive-ion HRESIMS (calcd. for C_15_H_21_O_4_^+^, 265.1434) that was appropriate for a molecular formula of C_15_H_20_O_4_. The NMR data of **5** showed that it shared the same trienedioic acid skeleton as epolactaene 4a [25]. The main difference was the absence of a methyl in **5**. The large vicinal coupling constant (*J* = 15.6 Hz) between H-6 (*δ*_H_ 6.23) and H-7 (*δ*_H_ 5.72) suggested an *E*-configuration for the ∆ ^6(7)^ double bond. In addition, the NOESY correlations between H_3_-1 (*δ*_H_ 1.65) and H-4 (*δ*_H_ 5.94), H-4 and H-6 and H_2_-9 (*δ*_H_ 2.45) and H_3_-15 (*δ*_H_ 1.74) indicated an *E*-configuration for the other double bonds (Figure 4). Therefore, the structure of **5**, named as steckfusarin F, was elucidated.

The other two known compounds were elucidated as (3*S*)-3,5-dimethyl-8-methoxy-3,4-dihydro-1*H*-isochromen-6-ol (**6**) [26] and 4-methyl-5,6-dihydro-2*H*-pyran-2-one (**7**) [27] by comparing their NMR and MS data with the data reported in the literature.

The new compounds (**1**–**5**) were evaluated for antioxidant, antibacterial, antifungal, antiviral, cytotoxic, anti-inflammatory, cholesterol-lowering and PFKFB3 and PI3K kinase inhibitory activities in vitro. 

The isolated compounds (**1**–**5**) were evaluated for their cytotoxic activities against twenty human cancer cell lines (A549, MKN-45, HCT 116, HeLa, K-562, MCF7, HepG2, SF126, DU145, CAL-62, 786-O, TE-1, 5637, GBC-SD, L-02, PATU8988T, HOS, A-375, A-673 and 293T) via CCK-8 (Dojindo) method. However, none of the compounds showed obvious cytotoxicity. Likewise, none of the new compounds did exhibited any growth inhibition when tested against *Klebsiella pneumonia*, *Candida albicans Berkhout*, *Staphyloccocus aureus*, *Colletotrichum gloeosporioiles*, *Magnaporthe grisea* and *Clerotinia miyabeana Hanzawa*. In addition, none of the new compounds exhibited cholesterol-lowering or PFKFB3 and PI3K kinase inhibitory activities. Compound **1** showed weak inhibitory activity against lipopolysaccharide-inducted nitric oxide (NO) in RAW 264.7 cells at a concentration of 20 µM. 

The radical scavenging activities of all the isolated compounds were tested against DPPH; compounds **1**, **2** and **4** showed radical scavenging activity against DPPH, with 37.3%, 54.3% and 45.8% inhibition at concentrations of 100 µg/mL, respectively. Due to the low mass of the active compounds, only compound **1** was measured for IC_50_. It had an IC_50_ value of 74.5 µg/mL, whereas ascorbic acid, used as a positive control, had an IC_50_ value of 7.5 µg/mL. Furthermore, molecular docking analyses between the active compounds and superoxide dismutase (PDB ID: 7wx0) were performed to gain functional and structural insights (Figure 5). The results showed that compound **1** could interact with superoxide dismutase at the entrance of the catalytic pocket, with a calculated binding affinity of −6.3 kcal/mol. Compound **1** interacted with the residues SER-32 and GLN-153 via two hydrogen bonds, and five hydrophobic interactions were formed between compound **1** and the residues LYS-4, VAL-6, HIS-20, ALA-152 and GLN-153. The active site of **1** contained methoxy and carboxyl groups. The molecular docking results demonstrated that the negative binding free energy value between compound **2** and superoxide dismutase was −6.6 kcal/mol. Interaction modes revealed six hydrogen bonding interactions with the residues GLN-23, ARG-79, ARG-79, SER-102, LEU-103 and ILE-104. In addition, there were two hydrophobic interactions with LEU-103 and ALA-105 in the active site of superoxide dismutase. The active site of **2** contained hydroxyl and carboxyl groups and an epoxy ring. The size of compound **1** was 51 × 51 × 47, centered at *x*: 17.894, *y*: 7.9, *z*: 82.091. The size of compound **2** was 49 × 44 × 49, centered at *x*: 17.894, *y*: 7.9, *z*: 82.091. Compounds **1** and **2** had no π-π stacking and π-cation interactions. The docking studies suggested that compounds **1** and **2** could inhibit superoxide dismutase by tightly binding to catalytic amino acid residues through different types of interactions.

Although many activities of fusarin derivatives have been reported [4,8,13,17], no obvious activities were detected for compounds **1**–**5**. It was reported that the tetrahydropyran ring and methylation of the acid group from the sidechain were crucial to the activity of these compounds [21]. However, none of the new compounds were methylated at C-19, which may be the reason why they did not display obvious activities.

## 3. Materials and Methods

### 3.1. General Experimental Procedures

Optical rotations were recorded using a Perkine Elmer 341 polarimeter (Hertford, UK). ECD spectra were recorded with a Chirascan circular dichroism spectrometer (Applied Photophysics, Surrey, UK). UV spectra were recorded on a UV-2600 UV–vis spectrophotometer (Shimadzu, Kyoto, Japan). The 1D and 2D NMR spectra were recorded on a Bruker AC 500 and 700 NMR (Broker, Fallanden, Switzerland) spectrometer with TMS as the internal standard. HRESIMS spectra were measured with a Bruker micro TOF-QII (Bruker, Fallanden, Switzerland) mass spectrometer in positive/negative ion mode. Silica gel GF-254 (10–40 mm) was used for thin-layer chromatography (TLC) (Qingdao Marine Chemical Factory, Qingdao, China). Sephadex LH-20 (Amersham Biosciences, Uppsala, Sweden) and silica gel (200–300 mesh, 100–200 mesh) (Qingdao Marine Chemical Factory, Qingdao, China) were applied in column chromatography (CC). HPLC was carried out on a Hitachi Primaide with a YMC ODS Series column (YMC-Pack ODS-A, YMC Co. Ltd. (Kyoto, Japan), 250 × 10 mm i.d., S-5 μm, 12 nm). All solvents were analytical grade (Tianjin Fuyu Chemical and Industry Factory). The fermentation culture medium and reagents were obtained from Guangzhou Haili Aquarium Technology Company, Guangzhou, China. 

### 3.2. Fungal Strain

The fungal strain *Penicillium steckii* SCSIO 41040 was isolated from a green algae *Botryocladia* sp. that was collected from the South China Sea. The isolated fungal strain was stored on Muller Hinton broth (MB) agar (malt extract 16 g, agar 18 g, sea-salt 30 g, water 1 L and pH 7.4–7.8) slants at 4 °C and then deposited in CAS Key Laboratory of Tropical Marine Bio-resources and Ecology, South China Sea Institute of Oceanology, Chinese Academy of Sciences, Guangzhou, PR China. The ITS1-5.8S-ITS2 sequence region (550 base pairs (bp), GenBank accession no. OP349656) of strain SCSIO 41040 was amplified via the PCR process. DNA sequencing showed that it shared significant homology (100%) with *Penicillium steckii* (GenBank accession no. NR111488). 

### 3.3. Fermentation and Extraction

A few loops of cells of the strain SCSIO 41040 were inoculated into a 500 mL Erlenmeyer flask containing 150 mL of seed medium (malt extract 1%, yeast extract 0.4%, glucose 0.4% and pH 7.2) and cultivated on a rotary shaker at 180 rpm and 28 °C for 48 h as a seed culture. Then, a large-scale fermentation of the fungal strain SCSIO 41040 was incubated at 25 °C in 1 L conical flasks containing a solid medium (300 mL/flask) composed of 200 g rice and 220 mL 3.2% (NaCl 3.2 g/H_2_O 100 mL) artificial seawater. After 32 days, the fermented material from each flask was extracted successively with EtOAc (700 mL/flask) and the combined EtOAc extract was suspended in MeOH and extracted with petroleum ether to remove rice oil. Finally, the MeOH solution was concentrated under reduced pressure to obtain a reddish-brown extract (56.0 g).

### 3.4. Isolation and Purification

The extract was subjected to silica gel column chromatography (CC), eluting with a gradient CH_2_Cl_2_-MeOH (100:0–0:100, *v*/*v*) to give 15 fractions based on TLC properties. Fr.3 was separated by semipreparative HPLC (40% MeCN/H_2_O + 0.3% TFA, 2.0 mL/min) to provide **6** (10.7 mg, *t*_R_ 17 min) and **7** (9.2 mg, *t*_R_ 8 min). Fr.5 was separated into 8 subfractions (Fr.5.1–5.8) using ODS silica gel chromatography via elution with MeOH/H_2_O (5–100%). Fr.5.6 was directly separated via semipreparative HPLC (56% MeOH/H_2_O + 0.3% TFA, 2.7 mL/min) to offer **4** (4.5 mg, *t*_R_ 44 min). Fr.5.7 was separated by semipreparative HPLC (35% MeOH/H_2_O + 0.3% TFA, 2.7 mL/min) to give **5** (8.5 mg, *t*_R_ 37 min). Fr.6 was applied to ODS silica gel chromatography to give eight fractions (Fr.6.1–6.8). Fr.6.2 and Fr.6.3 were further divided into 6 (Fr.6.2.1–Fr.6.2.6) and 11 (Fr.6.3.1–Fr.6.3.11) parts through a Sephadex LH-20 with MeOH and semipreparative HPLC (56% MeOH/H_2_O + 0.3% TFA, 2.5 mL/min), respectively. Fr.6.2.2 was purified by semipreparative HPLC (58% MeOH/H_2_O + 0.3% TFA, 2.7 mL/min) to yield **3** (3.0 mg, *t*_R_ 15 min). Fr.6.2.6 was separated by semipreparative HPLC (37% MeCN/H_2_O + 0.3% TFA, 3.0 mL/min) to yield **1** (23.2 mg, *t*_R_ 42 min). Fr.6.3.10 was directly separated via semipreparative HPLC (34% MeCN/H_2_O + 0.3% TFA, 2.5 mL/min) to provide **2** (5.8 mg, *t*_R_ 33 min).

Steckfusarin A (**1**): yellow oil; [α]D25 − 14.8°; UV (MeOH) *λ*_max_ (log *ε*) 235 (3.94) nm; ^1^H and ^13^C NMR data, Table 1; HRESIMS *m*/*z* 390.1912 [M + H]^+^ (calcd for C_21_H_28_NO_6_^+^, 389.1911).

Steckfusarin B (**2**): yellow oil; [α]D25 + 24.4°; UV (MeOH) *λ*_max_ (log *ε*) 213 (4.27), 233 (4.27) nm; ^1^H and ^13^C NMR data, Table 1; HRESIMS *m*/*z* 347.1609 [M − H]^−^ (calcd for C_20_H_24_NO_6_^−^, 374.1609).

Steckfusarin C (**3**): pink oil; [α]D25 − 1.7°; UV (MeOH) *λ*_max_ (log *ε*) 212 (4.03), 232 (4.01) nm; ^1^H and ^13^C NMR data, Table 1 and Table 2; HRESIMS *m*/*z* 476.2290 [M + H]^+^ (calcd for C_25_H_34_NO_8_^+^, 476.2279).

Steckfusarin D (**4**): yellow oil; [α]D25 – 8.5°; UV (MeOH) *λ*_max_ (log *ε*) 204 (4.21) nm; ^1^H and ^13^C NMR data, Table 2; HRESIMS *m*/*z* 346.2016 [M + H]^+^ (calcd for C_20_H_28_NO_4_^+^, 346.2013).

Steckfusarin E (**5**): yellow oil; UV (MeOH) *λ*_max_ (log *ε*) 216 (4.14) nm; ^1^H and ^13^C NMR data, Table 1; HRESIMS *m*/*z* 265.1431 [M + H]^+^ (calcd for C_15_H_21_O_4_^+^, 265.1434).

### 3.5. ECD Calculation

The relative configurations of the new compounds were subjected to random conformational searches using Spartan’14 and Gaussian 09 software with the Merck molecular force field (MMFF) and density functional theory (DFT)/TDDFT, respectively. The MMFF conformational search produced low-energy conformers with a Boltzmann population of more than 5%, which were geometrically optimized using the DFT method at the B3LYP/6-311G* level in MeOH using the IEFPCM model. For the stable conformers of the new compounds, the overall theoretical calculation of ECD was achieved in MeOH using time-dependent density functional theory at the B3LYP/6-311G* level. We used the 0.2–0.4 eV half bandwidth Multiwfn to generate the ECD spectra of different conformations and calculated the contribution according to the Boltzmann of each conformation after the UV correction [28].

### 3.6. Cell Culture and Cytotoxic Bioassay

The obtained fusarin analogues (**1**–**5**) were evaluated for their cytotoxic activities against twenty cell lines, A549, MKN-45, HCT 116, HeLa, K-562, MCF7, HepG2, SF126, DU145, CAL-62, 786-O, TE-1, 5637, GBC-SD, L-02, PATU8988T, HOS, A-375, A-673 and 293T (Shanghai Cell Bank, Chinese Academy of Sciences). The cytotoxic activity was determined via the CCK-8 (Dojindo) method [29].

### 3.7. Antimicrobial Assay

All new compounds were tested for antibacterial activity against *Klebsiella pneumonia*, *Candida albicans Berkhout*, *Staphyloccocus aureus*, *Colletotrichum gloeosporioiles*, *Magnaporthe grisea* and *Clerotinia miyabeana Hanzawa*. The tests were performed in 96-well plates using a modification of the broth microdilution method [30].

### 3.8. Anti-Inflammatory Assay

All new compounds were evaluated for their inhibitory activity against LPS-induced NF-κB activation in RAW264.7 cells using a luciferase reporter gene assay [31]. The RAW264.7 cells, stably transfected with the NF-κB luciferase reporter gene, were plated in triplicate into 96-well plates for all treatments and controls. The compounds (20 µM) and BAY11-7082 (an NF-κB inhibitor used as a positive control (5 µM); Sigma-Aldrich, St. Louis, MO, USA) were used to pretreat the cells for 30 min, followed by stimulation with 5 µg/mL LPS for 8 h. The cells were harvested and luciferase activity measured using a luciferase assay system (Promega, Madison, WI, USA).

### 3.9. Cholesterol Transport Mechanism

Pancreatic triglyceride lipase (PTL) and Niemann-Pick C1-like 1 (NPC1L1) were the crucial targets involved in cholesterol cellar uptake. The inhibitory activity of the compounds against PTL was evaluated by colorimetry. Surface plasmon resonance (SPR) was used to analyze the binding of the compounds with NPC1L1, and the activities of targeted compounds for NPC1L1 were studied [32,33].

### 3.10. PFKFB3 Kinase Inhibitory Activity

PFKFB3 kinase inhibitory activity was measured using an ADP-Glo Kinase Assay kit (Promega) according to a published modified protocol [34]. Compound solution (1 μL, with a final concentration of 20 μM in 1% DMSO) and 2 μL enzyme solution were added to 384-well plates, followed by incubation at room temperature (RT) for 30 min. After 2 h incubation at RT, 4 μL ADP-Glo Reagent was added and then the incubation continued for 1 h prior to the addition of 8 μL Kinase Detection Reagent and a further incubation of 1 h. The luminous signal was measured using an Envision flat panel reader (PerkinElmer, Waltham, MA, USA).

### 3.11. PI3K Kinase Inhibitory Activity

An amount of 0.5 μL of the compounds was preincubated with 14.5 μL of enzyme and PIP2 substrate for 10 min before addition of 5 μL of ATP to achieve a final ATP concentration of 10 μM [35]. The total reaction volume was 20 μL, and the reaction was allowed to proceed for 45 min at room temperature before the addition of the stop solution and detection mixture provided in the kit. The luminous signal was measured using an Envision flat panel reader (PerkinElmer).

### 3.12. Measurement of AChE Inhibition Activity

We evaluated the AChE enzyme inhibitory activity of compounds according to the slightly modified spectrophotometric method [36]. Tacrine was used as a positive control.

### 3.13. Molecular Docking Analysis

The crystal structure of neuraminidase (PDB ID: 7wx0) was retrieved from the Protein DataBank. We used AutoDockTools (Version 1.5.7) to conduct the molecular docking. The structures were generated in ChemBio3D Ultra 14.0 (ChemBioOffice version 14.0), followed by an MM2 calculation to minimize the conformation energy. The docking pose that had the lowest binding energy was represented as the most favorable binding conformation.

## 4. Conclusions

In this study, five new fusarin derivatives were obtained from the marine algicolous fungus *Penicillium steckii* SCSIO 41040, which had been fermented using rice solid medium. The new structures, including absolute configurations, were determined using spectroscopic methods coupled with the calculated ECD. Bioassays were used to screen the antioxidant, antibacterial, antifungal, antiviral, cytotoxic, anti-inflammatory, cholesterol-lowering and PFKFB3 and PI3K kinase inhibitory activities. Compound **1** exhibited antioxidant activity against DPPH, with an IC_50_ value of 74.5 µg/mL. In addition, compound **1** showed weak anti-inflammatory activity at a concentration of 20 µM. Compared with the previously reported fusarin derivatives [10,14,19,23], compounds **1**–**5** have no obvious antitumor activities, possibly because the new compounds lacked methylation at C-19.

## Figures and Tables

**Figure 1 marinedrugs-21-00532-f001:**
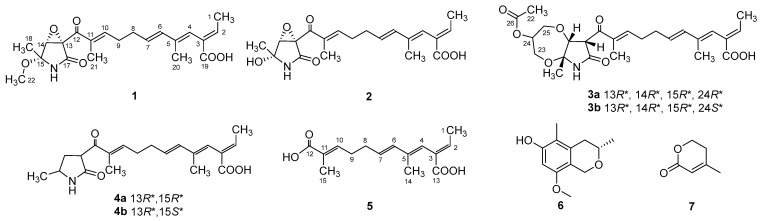
Chemical structures of compounds **1**–**7**.

**Figure 2 marinedrugs-21-00532-f002:**
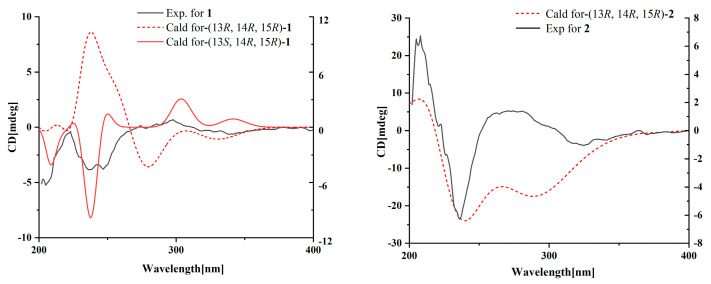
Experimental and calculated ECD spectra for compounds **1** and **2**.

**Figure 3 marinedrugs-21-00532-f003:**
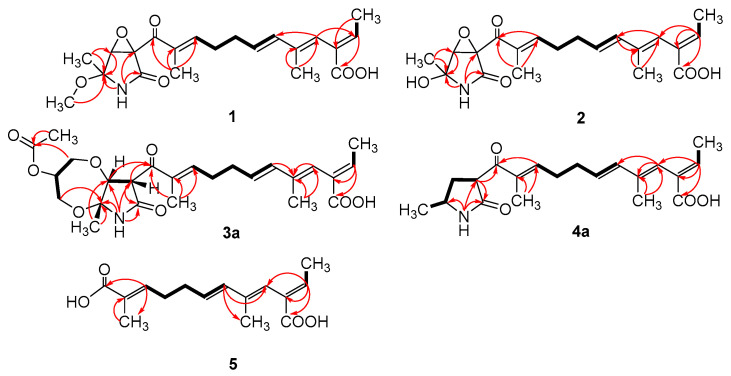
Key COSY (bold lines) and HMBC (arrows) correlations of compounds **1**–**5**.

**Figure 4 marinedrugs-21-00532-f004:**
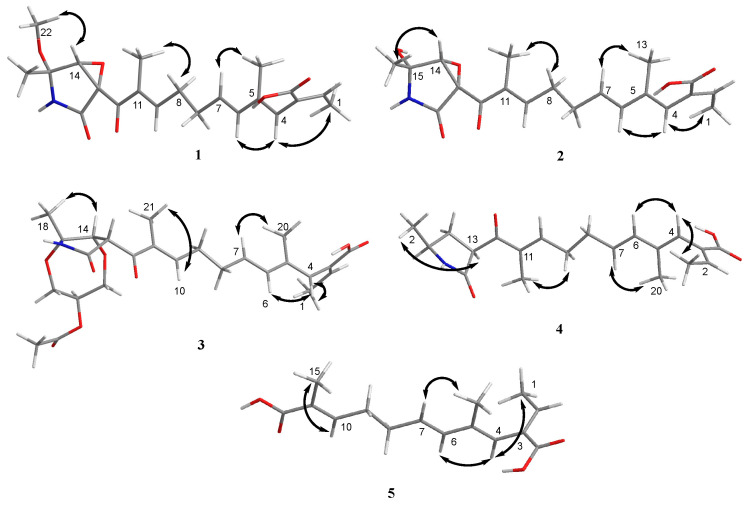
Key NOESY (double arrow) correlations of compounds **1**–**5**.

**Figure 5 marinedrugs-21-00532-f005:**
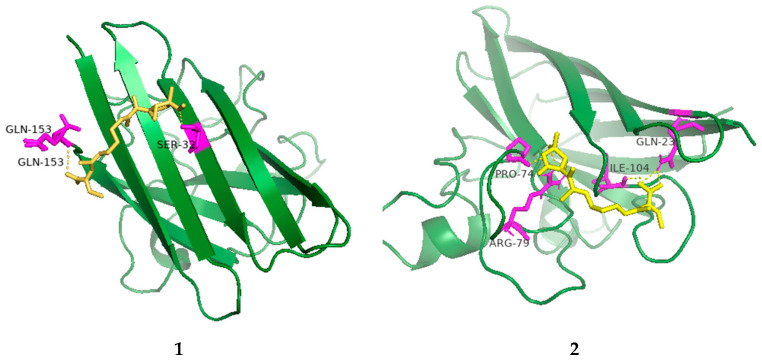
Low-energy binding conformations of **1** and **2** bound to superoxide dismutase (generated by molecular docking).

**Table 1 marinedrugs-21-00532-t001:** ^1^H and ^13^C NMR spectroscopic data for compounds **1**–**3** and **5** (700, 175 MHz, DMSO-*d*_6_).

No.	1		2		3		5	
	*δ*_C_, Type	*δ*_H_ (*J* in Hz)	*δ*_C_, Type	*δ*_H_ (*J* in Hz)	*δ*_C_, Type	*δ*_H_ (*J* in Hz)	*δ*_C_, Type	*δ*_H_ (*J* in Hz)
1	15.6 CH_3_	1.66 (d, 7.3)	15.6 CH_3_	1.65 (d, 7.1)	14.5 CH_3_	1.64 (dd, 7.2, 0.9)	15.6 CH_3_	1.65 (d, 7.1)
2	138.5 CH	6.80 (q, 7.3)	138.4 CH	6.80 (q, 7.1)	138.8 CH	6.78 (q, 7.2)	138.1 CH	6.78 (q, 7.1)
3	131.0 C		131.0 C		131.1 C		131.3 C	
4	123.5 CH	5.94 s	123.5 CH	5.93 s	123.5 CH	5.92 s	123.6 CH	5.94 s
5	136.7 C		136.7 C		137.0 C		136.5 C	
6	134.9 CH	6.25 (d, 15.6)	134.9 CH	6.25 (d, 15.6)	134.7 CH	6.26 (dd, 15.6, 7.2)	134.5 CH	6.23 (d, 15.6)
7	128.4 CH	5.73 (dt, 15.6, 7.0)	128.5 CH	5.74 (dt, 15.6, 7.0)	129.2 CH	5.79 (dt, 15.6, 7.0)	128.3 CH	5.72 (dt, 15.6, 7.0)
8	40.0 CH_2_	2.26 (q, 7.2)	31.0 CH_2_	2.25 (q, 7.3)	31.3 CH_2_	2.32 m	28.1 CH_2_	2.26 m
9	28.7 CH_2_	2.42 (q, 7.2)	28.9 CH_2_	2.40 (q, 7.3)	29.4 CH_2_	2.40 m	31.3 CH_2_	2.45 m
10	148.0 CH	6.86 (td, 6.8, 0.7)	147.5 CH	6.92 (dt, 7.2, 1.6)	147.4 CH	6.93 (td, 6.8, 0.9)	140.6 CH	6.66 (td, 6.9, 0.7)
11	135.5 C		135.4 C		137.5 C		128.9 C	
12	190.0 C		190.2 C		197.0 C		169.0 C	
13	62.2 C		62.6 C		48.8 CH	4.93 (d, 8.4)	168.1 C	
14	64.4 CH	4.28 (d, 2.3)	64.1 CH	4.1 (d, 2.3)	77.2 CH	4.33 (d, 8.4)	14.3 CH_3_	1.55 s
15	87.4 C		83.0 C		82.9 C		12.4 CH_3_	1.74 s
16-NH		8.82 s		8.63 s		8.89 s		
17	169.9 C		169.4 C		170.5 C			
18	18.7 CH_3_	1.39 s	22.3 CH_3_	1.39 s	20.0 CH_3_	1.48 s		
19	167.9 C		168.0 C		168.2 C			
20	14.2 CH_3_	1.56 s	14.3 CH_3_	1.55 s	15.8 CH_3_	1.54 s		
21	10.8 CH_3_	1.75 s	10.9 CH_3_	1.75 s	11.5 CH	1.72 s		
22	49.5 CH_3_	3.26 s						

**Table 2 marinedrugs-21-00532-t002:** Partial ^1^H and ^13^C NMR spectroscopic data for **3** (700, 175 MHz, DMSO-*d*_6_).

No.	3a		3b	
	*δ*_C_, Type	*δ*_H_ (*J* in Hz)	*δ*_C_, Type	*δ*_H_ (*J* in Hz)
22	20.7 CH_3_	1.98 s	21.0 CH_3_	2.00 s
23	61.2 CH_2_	3.51 m	62.8 CH_2_	3.34 m
		3.67 m		3.99 m
24	65.6 CH	4.16 m	69.4 CH	3.63 m
25	63.1 CH_2_	3.34 m	65.9 CH_2_	3.87 m
		3.99 m		4.02 m
26	170.8 C		170.9 C	

**Table 3 marinedrugs-21-00532-t003:** ^1^H and ^13^C NMR spectroscopic data for **4** (700, 175 MHz, DMSO-*d*_6_).

No.	4a		4b	
	*δ*_C_, Type	*δ*_H_ (*J* in Hz)	*δ*_C_, Type	*δ*_H_ (*J* in Hz)
1	15.7 CH_3_	1.64 (d, 7.1)	15.7 CH_3_	1.64 (d, 7.1)
2	138.4 CH	6.76 (q, 7.1)	138.4 CH	6.76 (q, 7.1)
3	131.5 C		131.5 C	
4	123.8 CH	5.95 s	123.8 CH	5.95 s
5	136.2 C		136.4 C	
6	134.7 CH	6.26 (d, 15.6)	134.7 CH	6.26 (d, 15.6)
7	128.8 CH	5.78 (dt, 15.6, 7.0)	128.8 CH	5.78 (dt, 15.6, 7.0)
8	31.2 CH_2_	2.30 m	31.2 CH_2_	2.30 m
9	29.0 CH_2_	2.38 (q, 7.3)	29.0 CH_2_	2.38 (q, 7.3)
10	145.4 CH	6.85 (td, 7.2, 0.7)	145.8 CH	6.91 (td, 7.2, 0.7)
11	136.4 C		136.6 C	
12	198.3 C		198.4 C	
13	47.8 CH	4.30 (q, 9.1)	48.9 CH	4.33 (q, 9.1)
14	33.0 CH_2_	1.72 m	33.3 CH_2_	1.80 m
		2.27 m		2.33 m
15	47.3 CH	3.60 (q, 6.6)	48.4 CH	3.65 (q, 6.6)
16-NH		7.90 s		7.90 s
17	172.5 C		172.8 C	
18	22.1 CH_3_	1.10 (d, 6.2)	22.3 CH_3_	1.12 (d, 6.2)
19	168.2 C		168.2 C	
20	14.3 CH_3_	1.56 s	14.3 CH_3_	1.56 s
21	11.4 CH_3_	1.72 s	11.5 CH_3_	1.72 s

## Data Availability

Not applicable.

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
