# Peer review of "New Fusarin Derivatives from the Marine Algicolous Fungus Penicillium steckii SCSIO41040"

_marinedrugs, 2023, doi:10.3390/md21100532_

Round 1

Reviewer 1 Report

In the manuscript by Song et al. the authors describe the elucidation of the structures of 5 new fusarin derivates and their testing for biological activity for all of them. Interestingly no activities were found, and the authors try to understand that result, comparing with the substitution pattern of other fusarins. 

The presentation is coherent and well done. It would be nice if the experimental data could be deposited digitally, including the molecules in NMReDATA format. 

I believe that there is a typo in line 176, where the IC50 values of compound 1 and ascorbic acid are compared, proabalby the 74.5 ug/ml should read 7.45 ug/ml. 

Author Response

(1) The presentation is coherent and well done. It would be nice if the experimental data could be deposited digitally, including the molecules in NMReDATA format.

Reply: We gratefully appreciate for your comments. We uploaded the NMR spectra to the Harvard dataverse, and the DOI number (https://doi.org/10.7910/DVN/PXCKSJ) was placed in the supporting information document.

(2) I believe that there is a typo in line 176, where the IC50 values of compound 1 and ascorbic acid are compared, proabalby the 74.5 ug/ml should read 7.45 ug/ml.

Reply: The IC50 value of compound 1 was 74.5 ug/ml. We just want to list the IC50 value of the positive drug, and we have adjusted for the misrepresentation of the sentence.

Reviewer 2 Report

Dear Authors,

In this study, the authors isolated and structurally identified five new fusarin derivatives from the marine fungus Penicillium steckii. The methodology is thoroughly explained and the results are clearly presented.  The discussion is well conducted and the conclusions are accurate. Supplementary material with detailed NMR data supports this manuscript.

In addition, biological antioxidants and anti-inflammatory properties were investigated. I propose to accept this manuscript after checking English and references.

Kind regards,

Please check English carefully

Author Response

In this study, the authors isolated and structurally identified five new fusarin derivatives from the marine fungus Penicillium steckii. The methodology is thoroughly explained and the results are clearly presented. The discussion is well conducted and the conclusions are accurate. Supplementary material with detailed NMR data supports this manuscript.

In addition, biological antioxidants and anti-inflammatory properties were investigated. I propose to accept this manuscript after checking English and references.

Kind regards

Reply: Thank you for your comments and encouragement.

Reviewer 3 Report

In this manuscript, authors carried out a chemical investigation on the marine algicolous fungus Penicillium steckii SCSIO 41040, which lead to the discovery of five new fusarin derivatives, steckfusarins A−E (15), along with two known natural products 6 and 7. Their structures were established by the extensive analysis of NMR data. Furthermore, the absolute configurations of compounds 1 and 2 were determined by ECD calculations. Additionally, these compounds were subjected to an array of bioassays. The results revealed compounds 1 and 2 exhibited radical scavenging activity against DPPH, and molecular docking study disclosed the binding modes between the two active compounds and superoxide dismutase. Moreover, compound 1 showed weak anti-inflammatory activity. These findings were important, which made this work worth publishing in this journal.

However, revisions were required.

1. It was said that fusarins were usually isolated from the genus Penicillium. Were there any fusarins reported from the fungus Penicillium steckii?

2. The expression of structure elucidation for compound 1 could be improved. Accompanied with the establishment of the 2-pyrrolidone ring, the position of epoxy ring and the substitution of methyl and methoxyl groups could be figured out. And please reconsider the HMBC correlation from H3-21 to C-17. Probably, it was a dummy signal in Figure S6.

3. The cofacial orientation of CH3-18/H-14 was determined, but the relative configuration of C-13 in compound 1 was not mentioned. As shown in Table S1 and S2, the conformer 13R,14R,15R-1 was used for the ECD calculations. Due to the uncertainty of the configuration of C-13, it is better to conduct additional calculations for the conformer 13S,14R,15R-1.

4. Since compounds 1 and 2 differed at the replacement of the methoxyl by the hydroxyl, it is better to use compound 1 for comparison to promote the structure elucidation of compound 2.

5. As the chemical structure of dothilactaene B shown in the reference ‘Molecules, 2021, 26, 59’, compound 3 was different from dothilactaene B by the presence of acetoxyl group at C-24 and the absence of methyl ester. Indeed, both of them had CH3-18. Please check and revise the corresponding sentences.

6. As shown in Figure 1, compounds 4 and 2 differed at the hydroxyl group and an epoxy system at 2-pyrrolidone ring structure.

7. Tables: The splitting patterns of H-2 of new compounds 25 should be checked. (BYW, the chemical shift of H-2 was missing in Table 1.) As H-2 was adjacent to CH3-1, the splitting pattern of H-2 usually was q not d. And the 1H NMR spectra of these new compounds showed q for H-2. Please also check other data, such H-2 of compound 2 in Table 1.

Other revisions:

1. P2L49: ‘calcd for, 389.1911’ → ‘calcd for C21H28NO6+, 390.1911’

2. P2L54: ‘one methine’ → ‘one oxygenated methine’

3. P2L54: ‘eight carbonyls’ → ‘eight quaternary carbons’

4. P2L61: ‘2D NMR spectrum’ → ‘2D NMR spectra’

5. Please add the chemical shifts of NH of the new compounds in the Tables.

6. P2L71: ‘m/z 347.1609 [M − H],  calcd for, 374.1609)’ → ‘m/z 374.1609 [M − H], calcd for C20H24NO6, 374.1609’

7. P3L99: ‘calcd for, 476.2279’ → ‘calcd for C25H34NO8+, 476.2279’

8. Figure S25 caption: The compound number 3 was missing.

9. P3L105: ‘...2-pyrrolidone ring structure, a 8a-methyl-2-oxohexahydro...’ → ‘...2-pyrrolidone ring structure, which was connected via ether linkages to form a 8a-methyl-2-oxohexahydro...’

10. P5L148: As shown in Figure S47, the data ‘265.1413’ and chemical formula ‘C15H21NO4+’ should be revised as ‘265.1431’ and ‘C15H21O4+’.

11. P5L149: ‘C15H20NO4’ → ‘C15H20O4

12. P8L267: ‘C15H21NO4+’ → ‘C15H21O4+

13. Non-Italic for the text of the subsection 3.5. ECD Calculation, but Italics for the species names in the subsection 3.7. Antimicrobial Assay.

14. Please check the information for the references at the end of this manuscript, such as the pages for the references [10] and [15]. And update the information for the reference [28].

Author Response

(1) It was said that fusarins were usually isolated from the genus Penicillium. Were there any fusarins reported from the fungus Penicillium steckii?

Reply: We gratefully appreciate for your comments. There were no fusarins reported from the fungus Penicillium steckii.

(2) The expression of structure elucidation for compound 1 could be improved. Accompanied with the establishment of the 2-pyrrolidone ring, the position of epoxy ring and the substitution of methyl and methoxyl groups could be figured out. And please reconsider the HMBC correlation from H3-21 to C-17. Probably, it was a dummy signal in Figure S6.

Reply: We have improved the expression of structure elucidation for compound 1 in the first paragraph of page 2. And we agree with the experts that the HMBC correlation from H3-21 to C-17 was a dummy signal.

(3) The cofacial orientation of CH3-18/H-14 was determined, but the relative configuration of C-13 in compound 1 was not mentioned. As shown in Table S1 and S2, the conformer 13R,14R,15R-1 was used for the ECD calculations. Due to the uncertainty of the configuration of C-13, it is better to conduct additional calculations for the conformer 13S,14R,15R-1.

Reply: We have added calculated ECD spectrum of 13S,14R,15R-1 in Figure 2.

(4) Since compounds 1 and 2 differed at the replacement of the methoxyl by the hydroxyl, it is better to use compound 1 for comparison to promote the structure elucidation of compound 2.

Reply: Thanks for your suggestion. We illustrated the structure by comparing the structures of 2 and 1.

(5) As the chemical structure of dothilactaene B shown in the reference ‘Molecules, 2021, 26, 59’, compound 3 was different from dothilactaene B by the presence of acetoxyl group at C-24 and the absence of methyl ester. Indeed, both of them had CH3-18. Please check and revise the corresponding sentences.

Reply: Thank you for pointing out the shortcomings of the manuscript. We have checked and revised the corresponding sentences.

(6) As shown in Figure 1, compounds 4 and 2 differed at the hydroxyl group and an epoxy system at 2-pyrrolidone ring structure.

Reply: Thank you for the correction, we have revised it.

(7) Tables: The splitting patterns of H-2 of new compounds 25 should be checked. (BYW, the chemical shift of H-2 was missing in Table 1.) As H-2 was adjacent to CH3-1, the splitting pattern of H-2 usually was q not d. And the 1H NMR spectra of these new compounds showed q for H-2. Please also check other data, such H-2 of compound 2 in Table 1.

Reply: We have checked all the data, and made corrections.

Other revisions:

(1) P2L49: ‘calcd for, 389.1911’ → ‘calcd for C21H28NO6+, 390.1911’

Reply: Thank you for your correction.

(2) P2L54: ‘one methine’ → ‘one oxygenated methine’

Reply: Thank you for your comments.

(3) P2L54: ‘eight carbonyls’ → ‘eight quaternary carbons’

Reply: Thank you for your comments.

(4) P2L61: ‘2D NMR spectrum’ → ‘2D NMR spectra’

Reply: Thank you for your correction. We have fixed this error.

(5) Please add the chemical shifts of NH of the new compounds in the Tables.

Reply: The chemical shifts of NH of the new compounds have added in Tables 1 and 3, numbered 16-NH.

(6) P2L71: ‘m/ z 347.1609 [M − H], calcd for, 374.1609)’ → ‘m/z 374.1609 [M − H], calcd for C20H24NO6, 374.1609’

Reply: We have made corresponding corrections.

(7) P3L99: ‘calcd for, 476.2279’ → ‘calcd for C25H34NO8+, 476.2279’

Reply: We have made corresponding corrections.

(8) Figure S25 caption: The compound number 3 was missing.

Reply: Thank you for your careful read, the number 3 have been added.

(9) P3L105: ‘...2-pyrrolidone ring structure, a 8a-methyl-2-oxohexahydro...’ → ‘...2-pyrrolidone ring structure, which was connected via ether linkages to form a 8a-methyl-2-oxohexahydro...’

Reply: Thank you for your comments and revisions.

(10) P5L148: As shown in Figure S47, the data ‘265.1413’ and chemical formula ‘C15H21NO4+’ should be revised as ‘265.1431’ and ‘C15H21O4+

Reply: We appreciate your advice and we have fixed the errors.

(11) P5L149: ‘C15H20NO4’ → ‘C15H20O4

Reply: We have corrected this error.

(12) P5L267: ‘C15H20NO4’ → ‘C15H20O4

Reply: We have corrected this error.

(13) Non-Italic for the text of the subsection 3.5. ECD Calculation, but Italics for the species names in the subsection 3.7. Antimicrobial Assay.

Reply: The scientific names of the organisms in the manuscript were shown in italics. in the subsection 3.5. ECD Calculation, no scientific names were used, so there were no italics in this section.

(14) Please check the information for the references at the end of this manuscript, such as the pages for the references [10] and [15]. And update the information for the reference [28].

Reply: We have checked the references and corrected similar errors.

Reviewer 4 Report

New Fusarin Derivatives from the Marine Algicolous Fungus Penicillium steckii SCSIO41040

Yingying Song et al reported five new fusarin derivatives, steckfusarins A−E (1−5), and two known natural products (6, 7), were isolated and identified from the marine algicolous fungus Penicillium steckii SCSIO 41040. The new compounds including absolute configurations were determined by spectroscopic analyses and calculated electronic circular dichroism (ECD). All new compounds were evaluated for their antioxidant, antibacterial, antifungal, antiviral, cytotoxic, anti-inflammatory, antioxidant, cholesterol-lowering, acetyl cholinesterase (AChE) enzyme, 6-phosphofructo-2-kinase (PFKFB3) and phosphatidylinositol-3-kinase (PI3K) inhibitory activities. Biological evaluation results revealed that compound 1 exhibited radical scavenging activity against 2,2-diphenyl-1-picrylhydrazylhydrate (DPPH) with an IC50 value of 74.5 µg/mL. And compound 1 also showed the weak anti-inflammatory activity at the concentration of 20 µM.

This study was conducted clearly and could bring significant information for the readers. However, it is still more revise for better. Herein my comment in blow:

  1. The authors should provide 3D structure for present key NOESY of new compounds.
  2. Authors should provide evidence of the purity of new compounds. Because I see CD experiment, it seem not over 95 % purity.
  3. The discussion of new compounds needs more clearly for configuration.
  4. What is the statistic analysis of this study?
  5. Molecular docking analysis is very low, need more improve. What the active site and XYZ?

Minor comment:

(The ratio of solvent CH2Cl2-MeOH, 100:0-0:100, v/v), all for this manuscript.

Provide a detailed supplementary file

Minor editing of English language required

Author Response

(1) The authors should provide 3D structure for present key NOESY of new compounds.

Reply: Thanks for your suggestion. The 3D structures have added in Figure 4.(2) Authors should provide evidence of the purity of new compounds. Because I see CD experiment, it seems not over 95 % purity.

Reply: We re-analyzed the new compounds with HPLC, but due to the instability of compounds, the purity of the compounds could not be shown. Only the NMR and HRESIMS spectra in the supporting information document could be used as evidence of the purity of the compounds.

(2) The discussion of new compounds needs more clearly for configuration.

Reply: We added some details to describe the configuration of the new compounds.

(3) What is the statistic analysis of this study?

Reply: First, we have enriched the variety of fusarin derivatives. Secondly, through a large number of biological activity measurements, we inferred that the methylation of the acid group of the sidechain are essential for the activity of this compound.

(4) Molecular docking analysis is very low, need more improve. What the active site and XYZ?

Reply: We have tried multiple times to improve the docking effect. The active site of 1 were methoxy and carboxyl groups. The size of was 51 × 51 × 47, centered at x: 17.894, y: 7.9, z: 82.09. The active site of 2 were hydroxyl, carboxyl groups and epoxy ring. The size of compound 2 was 49 × 44 × 49, centered at x: 17.894, y: 7.9, z: 82.091.

Minor comment:

(The ratio of solvent CH2Cl2-MeOH, 100:0-0:100, v/v), all for this manuscript.

Reply: Volume ratios have been added in the text.

Round 2

Reviewer 3 Report

The authors nicely addressed all of my comments and revised accordingly. Only the splitting patterns of H-7 (dd or dt?) and H-10 (dt or td?) need to be checked over. Because these could be easily done, I recommend this manuscript to be accepted.

Author Response

Response to the Reviewer 3:

(1) The authors nicely addressed all of my comments and revised accordingly. Only the splitting patterns of H-7 (dd or dt?) and H-10 (dt or td?) need to be checked over. Because these could be easily done, I recommend this manuscript to be accepted.

Reply: Thank you for your careful read and comments. We have checked over only the splitting patterns of H-7 and H-10, and corrected it in Tables 1 and 3.